# New Therapeutic Options in Mild Moderate COVID-19 Outpatients

**DOI:** 10.3390/microorganisms10112131

**Published:** 2022-10-27

**Authors:** Claudio Ucciferri, Alessandro Di Gasbarro, Paola Borrelli, Marta Di Nicola, Jacopo Vecchiet, Katia Falasca

**Affiliations:** 1Clinic of Infectious Diseases, Department of Medicine and Science of Aging, University “G. d’Annunzio” Chieti-Pescara, 66100 Chieti, Italy; 2Laboratory of Biostatistics, Department of Medical, Oral and Biotechnological Sciences, University “G. d’Annunzio” Chieti-Pescara, 66100 Chieti, Italy

**Keywords:** pidotimod, immunomodulation, SARS-CoV2, therapy, safety, efficacy

## Abstract

Background: In recent years, the therapeutic options for COVID have significantly improved; however, the therapies are expensive with restricted access to drugs, and expeditious and difficult to manage at home. We investigated the effect of pidotimod in preventing hospitalization in patients with mild-moderate COVID-19. Methods: A total of 1231 patients between January and June 2021 were screened. A total of 184 patients with mild-moderate COVID-19 were enrolled and divided into two groups: group-A (97) had undergone therapy with pidotimod 800 mg bid for 7–10 days and group-B (87) had other therapies. We excluded those who had undergone complete vaccination course, monoclonal anti-spike/antivirals or the co-administration of pidotimod-steroid. The primary outcome chosen was the emergency room, hospitalization, and deaths for COVID-related causes; the secondary outcome chosen was the duration of COVID-19 illness. Results: A total of 34 patients (18.5%) required hospital treatment, 11 in group-A and 23 in group-B (11.3% vs. 26.4%, *p* = 0.008). The median disease duration in group-A was 21 days (IQR 17–27) vs. 23 (IQR 20–31) in group-B (*p* = 0.005). Patients in the pidotimod group had higher SpO2 in the walking test (IQR 96–99% vs. IQR 93–98%, *p* = 0.01) and a lower need for steroid rescue therapy (11.5% vs. 60.9%, *p* < 0.001). Conclusions: In the first phase of disease, pidotimod can represent an effective, low-cost, weapon, without restrictions of use, that is able to prevent a second aggressive phase and promote faster virological recovery.

## 1. Introduction

Severe acute respiratory syndrome coronavirus 2 (SARS-CoV-2) is responsible for COVID-19, a disease that in a few months become a pandemic starting at the end of 2019. As of 31 July 2022, the COVID-19 pandemic is still raging all around the world with more than 500 million cases and several million deaths. Despite more than 2 years since its start, SARS-CoV-2 infections continue to pose a public health threat in many countries [1].

Symptoms and signs of SARS-CoV-2 infection are not always specific and range from fever, dry cough, fatigue, headache, dysgeusia, anosmia, acute lung injury with shortness of breath to acute respiratory distress syndrome (ARDS), which can lead to patient death [2,3].

COVID-disease is classically divided into two pathogenic phases: the first is characterized by viral replication, the second, instead, is linked to the activation of an inflammatory response with the possible appearance of a cytokine storm. A rapid and well-coordinated innate immune response is the first line of defense against viral infections, but dysregulated and excessive immune responses may cause immunopathology [4,5].

Different therapeutic strategies have been adopted in order to moderate the cytokine storm in critical patients such as interleukin 1 (IL-1), interleukin 6 (IL-6) inhibitors [6] or Janus kinase inhibitor (JAK inhibitor) [7]. About 20% of patients present COVID with a disease that requires hospitalization and treatments on the above statements [8]. The aim of the most recent research is to try to reduce the hospitalization and therefore the mortality of patients. However, such options are expensive and are not always accessible and available. In light of this, it is necessary to develop less expensive strategies that allow for improvements in the outcome of COVID-19 patients, in order to allow them to be practicable for all patients. Pidotimod (3-L-pyroglutamyl-L-thiaziolidine-4 carboxylic acid), a synthetic dipeptide molecule with immunomodulatory properties [9], seems to be an ideal candidate [10].

In vivo and in vitro studies show that pidotimod’s immunomodulatory activity is focused on both adaptive and innate immunity: this dipeptide induces dendritic cell (DC) maturation, upregulates the expression of HLADR and co-stimulatory molecules, stimulates DCs to release pro-inflammatory molecules that drive T-cell proliferation and differentiation toward a Th1 phenotype, enhances natural killer (NK) cell functions, and promotes phagocytosis [11,12]. This mechanism seems to also play a role in COVID-19: recent studies have shown an anti-inflammatory effect on COVID-19 pneumonia [13] and a rapid rection in symptoms in paucisintomatic COVID patients when treated with pidotimod [14].

We investigated the effect of pidotimod in mild moderate COVID-19 outpatients. The aims of this study were as follows. The first objective was to evaluate the efficacy of pidotimod administration in COVID-19 infection in non-vaccinated subjects, in order to reduce the numbers in accessing the emergency room, hospitalization, and deaths. The secondary outcome was the number of days between the onset of disease and the date of the first negative swab.

## 2. Materials and Methods

*The study design* was by the Clinic of Infectious Diseases of the University ‘G. D’Annunzio’—PO SS Annunziata of Chieti, which enrolled patients by Special Units of Continuity of Care (USCA) for COVID-19 outpatients in Lanciano and Guardiagrele, Italy, between January and June 2021 with mild or moderate COVID-19. We considered patients from January to June 2021. All patients were proposed treatment by USCA based on different therapy options according to the local guidelines for home management of the COVID-19 patient redacted by the local health authority. The diagnosis was confirmed by PCR in a nasopharyngeal swab and mild/moderate disease was considered present when the basal oxygen saturation was >93%, there was no dyspnea at rest, and the modified early warning score (MEWS) was <6.

Exclusion criteria were: Severe/critical COVID-19 disease at time of treatment enrolment, treatment with anti-spike monoclonal antibodies, complete vaccination for SARS-CoV-2, incorrect pidotimod administration schedule (one daily dose, not on an empty stomach, incorrect dosage, treatment duration of less than 7 days) or timing (administration after the first 7 days of disease), co-administration of pidotimod and corticosteroids, and aged <18 years old.

The study was approved by the local institutional review board at the University “G. d’Annunzio” Chieti-Pescara, and all patients provided written informed consent to participate in the study.

### 2.1. Study Procedures

On admission, the demographic characteristics (age, gender), comorbidities (like hypertension, diabetes, obesity, COPD, etc.) and the history of COVID-19 disease (presence, type and onset of symptoms, date of positive swab, date of any close contact with infected subjects) were recorded; clinical data (respiration rate, peripherical oxygen saturation at rest and during the 6 min walking test, heart rate, blood pressure, body temperature, chest auscultation, ecografic lung ultrasound score, LUS) were collected during home medical examination and (partially) during daily telephone monitoring of the patient.

Pidotimod therapy consisted of administrations of 800 mg bid effervescent granules, to be taken on an empty stomach, for 10 days. The time indication for the administration of this molecule was within the first 7 days from the onset of the disease (identified as the earliest between the date of onset of symptoms and the date of first positive swab). For the control group, all patients who met the enrolment criteria who had not administered pidotimod or steroid were considered.

We considered the introduction of rescue therapy with corticosteroids in patients who, during the therapeutic course, experienced desaturation (SpO2 <94%), and therefore needed support with oxygen therapy at home, or in whom there was a persistent fever, resistant to any other therapy. When rescue therapy was introduced, pidotimod was discontinued.

### 2.2. Statistical Analysis

Descriptive analysis was carried out using median and interquartile range (IQR) for the quantitative variables and percentage values for the qualitative ones. The normality distribution for the quantitative variables was assessed by the Shapiro–Wilk test. The association between endpoint variables (outcome of ER access/hospitalizations and illness duration) and explicative variables was investigated by the Pearson χ2 test and non-parametric Wilcoxon rank-sum test for unpaired two-samples. Correlations among variables were tested using Spearman’s rho coefficients. Crude odds ratio (OR) and corresponding 95% CI were calculated in order to quantify the risk associated with the outcome of hospitalizations considered explicative variables using the Wald test. Multivariable logistic regression model was conducted to identify the mutually adjusted effect among the outcome of ER access/hospitalizations and the independent variables chosen on the basis of (1) the statistical significance (univariate analysis, *p* ≤ 0.05); and (2) the clinical judgment and their contribution to the model fit (likelihood-ratio test). The goodness of fit of the multivariable logistic regression model was assessed by the Hosmer–Lemeshow test. A multiple linear regression model was implemented to verify the relationship between the illness duration and the independent variables considered. Statistical significance was set at the level of ≤0.05. All analyses were performed using Stata software v17.1 (StataCorp, College Station, TX, USA).

## 3. Results

Of the 1231 patients managed by the USCA between January and June 2021, after the application of the exclusion criteria, 615 were enrolled and divided: 310 in the “pidotimod” group and 305 in the “not-pidotimod” group.

In the pidotimod group, 223 patients were excluded due to treatment discontinuation, incorrect dosage/duration of treatment, simultaneous administration of pidotimod and corticosteroids, and incomplete data. In the not-pidotimod group, 218 patients were excluded due to the incompleteness of the collected data. Finally, the study population consisted of 184 patients, 97 in the pidotimod group, and 87 in the non-pidotimod group (Figure 1).

The study population consisted of 46% male subjects with a median age of 53.0 (IQR 42.0–62.5) years. A total of 113 (65.7%) patients had at least one comorbidity. The median of disease duration was 22.0 (IQR 18.0–28.0), the BMI was 27.1 (IQR 23.8–29.4), and the median of fever duration was 3.0 (IQR 0.0–6.0).

The median number of unique symptoms developed during the disease course was 4.0 (IQR 3.0–6.0) symptoms (the incidence of the various symptoms is described in Table 1), the median SpO2 at enrolment was 98% (IQR 97.0–99.0), and the median minimum SpO2 during the course of disease was 96% (IQR 94.0–97.0). The median MEWS score at enrolment was 1 (IQR 0–3), and the median LUS at lung ultrasound at enrolment was 3 (IQR 0–6). The need for steroid use became necessary in 35% of the population (Table 1).

In the group treated with pidotimod (vs. who were treated with other therapy) there was a higher number of patients with comorbidities (59 vs. 54, *p* = 0.012); there was a greater number of symptoms during the course (median = 5.0 vs. median = 4.0, *p* < 0.001), with a predominance of mild symptoms (myalgia 70 vs. 50, *p* = 0.037; headache 50 vs. 30, *p* = 0.02), a higher minimum SpO2 [median = 96.0% (IQR 94.0–97.0) vs. median = 95.0% (IQR 93.0–97.0), *p* = 0.046], and a higher SpO2 in the 6-min walking test [median = 97.0% (IQR 96.0–99.0) vs. median = 97.0% (IQR 93.0–98.0), *p* = 0.01].

In this group, there was a lower need for rescue therapy with corticosteroids (11 (11.5%) vs. 53 (60.9%), *p* < 0.001), a lower need for ED access/hospitalization for COVID-related causes [11 (11.3%) vs. 23 (26.4%), *p* = 0.008], and shorter disease duration [median 21.0 days (IQR 17.0–27.0) vs. median = 23.0 days (IQR 20.0–31.0), *p* = 0.005] than in the group not treated with pidotimod (Table 2).

A total of 34 (18.48%) patients (52.9% female) needed hospital care: three were treated in the emergency room and then returned home, 31 were hospitalized and, of these, seven died. Hospitalized versus non-hospitalized patients were older [66 (IQR 54–79) vs. 50.5 (IQR 39–60) years old, *p* < 0.001], had a longer duration of illness [28 (IQR 23–38) vs. 22 (IQR 18–27) days, *p* < 0.01], a higher BMI [33.3 (IQR 27.1–40.8) vs. 26.2 (IQR 23.5–29.1), *p* < 0.02], frequently had more comorbidities (87.9% vs. 60.4% *p* < 0.01), more frequently had dyspnea (47.1% vs. 19.3% *p* < 0.001), a longer duration of fever [4 (IQR 2–7) vs. 3 (IQR 0–5) days, *p* < 0.05], a worse saturation at baseline [96% (IQR 95–98) vs. 98% (IQR 97–100), *p* < 0.001] and nadir [93% (IQR 90–94) vs. 96% (IQR 95–97), *p* < 0.001], a worse SpO2 at 6 min walking-test at nadir [93% (IQR 90.5–97) vs. 97% (IQR 95–99), *p* < 0.001], a higher MEWS at baseline [3 (IQR 1–5) vs. 0 (IQR 0–2), *p* < 0.001] and nadir [5 (IQR 3–7) vs. 1 (IQR 0–2), *p* < 0.001], worse LUS score at baseline [6 (IQR 2–10) vs. 2 (IQR 0–6), *p* < 0.001] and nadir [12 (IQR 8–14) vs. 4.5 (IQR 1.5–6.5), *p* < 0.001], the more frequently patients used steroids (50% vs. 31.5% *p* < 0.05), with earlier steroid use [7 (IQR 4.5–8) vs. 8,5 (IQR 8–11) day of disease, *p* < 0.001]. Instead, they used pidotimod less frequently (32.4% vs. 57.3% *p* < 0.01), and less often had headache (23.5% vs. 48% *p* < 0.01) and ageusia or anosmia (32.4% vs. 56.7% *p* < 0.01) as symptoms (Table 3).

The secondary outcome chosen in our study was disease duration. It has been observed that the longer duration of illness correlates with age (rho 0.184, *p* = 0.014), the presence of comorbidities [median = 23.0 (IQR 19.0–30.0) days vs. median = 20.5 (IQR 16.0–26.0) days, *p* = 0.026], a minor SpO2 at enrolment (rho −0.1831, *p* = 0.015), a minor minimum SpO2 during the course of illness (rho −0.266, *p* < 0.001), a minor SpO2 in the 6-min walking test (rho −0.1757, *p* = 0.030), and the need for access to the emergency room/hospitalization [median = 28.0 (IQR 23.0–38.0) days vs. median = 22.0 (IQR 18.0–27.0) days, *p* = 0.002]. The use of pidotimod, on the other hand, was correlated with a shorter illness duration [median = 21.0 (IQR 17.0–27.0) vs. median = 23.0 (IQR 20.0–31.0), *p* = 0.005]. No other statistically significant differences were detected.

Table 4 and Figure 2 show the results of the crude and adjusted ORs for ER access/hospitalization. Crude OR confirmed the association, as shown in Table 3, while the logistic regression model showed that the risk of ER access/hospitalization increased with age (OR 1.06, 95 CI% 1.03–1.09, *p* < 0.0001) and decreased in patients undergoing pidotimod therapy (OR 0.38, 95 CI% 0.16–0.89, *p* = 0.026).

The Hosmer–Lemeshow goodness-of-fit test indicates that the model describes the data well (χ^2^(129) = 107.36, *p* = 0.917).

Finally, the multiple linear model for the illness duration showed that as 1 year of age increased the duration of illness (*p* < 0.0001), it decreased for patients who had been given pidotimod therapy (*p* = 0.004) (Table 5).

## 4. Discussion

Since its beginning, the SARS-CoV-2 pandemic has put welfare systems under pressure, causing the proliferation of inhomogeneous and often very different home therapeutic strategies. A critical point in the home management of the COVID-19 patient was the lack of effective treatment strategies in the first week of illness.

The arrival of monoclonal antibodies (bamlanivimab-etesevimab, casirivimab-imdevimab, sotrovimab, etc.) and antivirals (molnupiravir, nirmatrelvir-ritonavir) has, in part, responded to this need by providing effective, but expensive and restricted access (from temporal and patient characteristics points of view), weapons [15,16,17,18]. Part of this arsenal has also been subject to a loss of efficacy linked to the emergence of new viral variants (for example, most of the available antibody combinations are currently unusable for this reason) [19,20,21,22,23]. We evaluated an alternative approach based on early moderation of the inflammatory response, directing it toward a more efficient, coordinated and appropriate response. On the basis of previous literature experiences, the drug that best suited this aim was pidotimod, a polypeptide with chemical and functional similarities to imiquimod and with immunomodulating properties, expressed through multiple actions on innate and acquired immunity.

Pidotimod favors the expression of TLR2 and TLR7 (over TLR4) in the respiratory tract with a correct antigenic presentation [20,24,25], stimulates the production of INF 1 and gamma with the establishment of the antiviral state and the predominance of a Th1 phenotype (over Th2 and Th17) [11,21,22] with inflammation control, and stimulates mucosal IgA production with a reduction in the amount of antigen available for ADE [26]. Finally, a population of M2-type alveolar macrophages is favored, which is found to be prevalent in subjects who do not develop severe symptoms [23,27,28].

The data that emerged from our study seems to be in agreement with what has been said thus far. In the pidotimod group, hospital access, hospitalizations, and deaths were lower with a statistically significant difference compared to the control group. These data could indicate a better control of the inflammatory response in these patients by an immune system that is adequately modulated and prepared, in the early stages of the infection, to give a better response. Consequently, less lung damage, less development of respiratory failure, ARDS, and finally, less need for care in a hospital setting would result.

Two other results that emerged from our study reinforce this hypothesis: the minimum SpO2 value found during the course of the disease and the SpO2 value found in the 6-min walking test were higher in patients in the pidotimod group. A higher SpO2 minimum value could indicate less cumulative lung injury in patients treated with pidotimod. The SpO2 value in the walking test is an index of the residual pulmonary respiratory reserve; the alteration in this index is earlier than the alteration of the SpO2 at rest in the case of lung damage. It is a pulmonary functional index, an indirect reflection of the state of damage to this organ [29,30]; then the difference between the two groups supports the hypothesis of a lower functional damage (probably linked to a lower anatomical damage) in the group treated with pidotimod.

This lower degree of lung inflammation is probably related to the reduced homing of inflammatory cells in this organ, also linked to the expression of Toll-like receptors 2 and 7, which correctly process viral antigens, and to the M2 phenotype of alveolar macrophages (producers of fewer chemokines and pro-inflammatory cytokines), both events stimulated by pidotimod [20,27].

Another finding in this direction is a minor need for rescue therapy with corticosteroids in the pidotimod group. The need to use corticosteroids indicates, on one hand, greater lung damage (reflected in a lower SpO2), and on the other hand, the presence of a more marked systemic inflammation (persistent fever), which, left to itself, probably would have caused lung damage and the need for hospital treatment (often the steroid was not able to reverse these processes): the early immunomodulation implemented by pidotimod might be able to prevent these advanced disease states, thus reducing the need to use corticosteroids as a last attempt of home therapy before hospital treatment. Symptomatologically, patients treated with pidotimod more often developed symptoms such as headache and myalgia, which could be part of less severe disease subsets, less associated with complications compared with subsets including respiratory symptoms and dyspnea (as emerged also in our analysis). From a clinical point of view, a better control of the inflammatory state could be reflected in a systemic symptomatology of minor importance and, in particular, in a lower percentage of relevant respiratory symptoms. In the pidotimod group, disease duration was also significantly shorter. This difference could be linked to the aforementioned result of a less severe disease with less need for hospitalization. An organized immune response, addressed by an early immunomodulation, could have produced a faster viral clearance: in particular, the stimulatory effect of pidotimod on the interferon 1 system could have earlier produced that so-called intracellular “antiviral state” necessary for the correct elimination of viral infections [22].

In our study, it emerged that the duration of the disease is directly correlated with the age of the patient and the presence of comorbidities, as is already known in the literature. It was highlighted how a clinically more severe disease is associated with a longer duration of positivity to SARS-CoV-2: this is probably linked to a more disorganized and harmful immune response that fails to obtain an efficient viral clearance. It can also be noted, with a value close to statistical significance (*p* = 0.06), how the use of the corticosteroid can be associated with a longer disease duration [31,32]: this could be linked to the more severe clinical picture that is usually present in patients who required the use of the steroid [33]. Additionally, in the sub-analysis of this secondary outcome, the role of pidotimod in reducing the number of days of positivity to SARS-CoV-2 emerged compared to patients who had undergone other therapies.

From the data of our study, it emerges that pidotimod, in a patient with mild/moderate COVID-19 and the right timing of the disease, can be an effective therapy in preventing the development of severe disease and the need for hospital care. At the time, we had different options for the viral replication phase (with stringent inclusion criteria and time frames) and for the inflammatory phase; we have no effective options to treat patients in the viral phase without risk factors, outside of the fifth day of disease, younger, or in therapy with drugs that interact with antivirals. All of these categories cannot access antivirals and pidotimod can represent a valid therapeutic option. Indeed, it is a low-cost alternative, compared to others that can be used in this setting, which can be used in almost any type of home patient without access restrictions and relevant drug interactions.

Another purpose of our work was to highlight a series of factors (clinical and non-clinical) related to the patient that could be associated with an unfavorable outcome or a longer duration of illness to provide additional patient assessment tools outside the hospital.

In addition to age, the presence of comorbidities and BMI, which are already known in the literature [34,35,36], other factors were significantly associated with a greater risk of hospital access: number of days with fever, presence of dyspnea, initial SpO2, SpO2 on walking test, MEWS score, LUS score, corticosteroid use, and day of illness on which they were introduced in therapy.

In patients with a worse outcome, the presence of fever was observed, on average, for a higher number of days. This could indicate a more pronounced and aggressive inflammatory process responsible for the lung damage, which then leads to hospitalization.

In this same direction goes the relief, in this group of patients, of a more frequent presence of dyspnea, evidence of lung damage already underway [37]. It could therefore be identified, within the multiform presentations of the COVID-19 disease, a subset with a greater risk of hospitalization characterized by fever and marked respiratory symptoms, which would deserve greater attention when it appears. On the other hand, symptoms such as ageusia, anosmia, headache, and myalgia could constitute a subset with a lower risk of nefarious evolution. For some of these, a protective effect has already been highlighted in the literature [38,39,40]. Patients who undergo hospitalization also showed fewer days without symptoms and a longer disease course, underlining the biggest burden of disease with a slower and longer recovery and therefore a longer positivity to SARS-CoV-2.

Although within the normal limits, saturation at rest was lower, right from enrolment, in patients who subsequently underwent hospitalization. The walking test was confirmed, even in our study, to be a useful tool capable of highlighting (even before SpO2 at rest) the presence of desaturation. An SpO2 below 94% during the walking test should be a wake-up call for any physician working with COVID patients outside the hospital.

MEWS is a clinical score, part of routine clinical practice from 1999, with the aim of the early identification of patients at high risk of admission to ICU; this has been validated for a general medical population and for a surgical population [41,42]. It is a quick score composed of parameters that can be easily collected and therefore is applicable at home, by the bedside. In previous studies on COVID-19 patients, this score has not been shown to be able to predict the evolution toward severe forms of the disease [43]; this is probably linked to its late use (application in Emergency Room). In our study, this score was also useful in predicting hospital admission for COVID-19 patients, identifying a value greater than 2 as deserving particular attention.

In our study, we identified a marked difference in the ultrasound findings of those who were hospitalized and those who did not need.

LUS is a validated ultrasound score based on the “ring-down” artifact that indicates the presence of fluid inside the lung parenchyma (edema, alveolar or interstitial), which is directly proportional to the severity of the lung inflammation [44,45]. Lung ultrasound is shown to be an adequate tool to identify early deterioration of the lung parenchyma. Previous studies have attempted to identify a minimum LUS value to indicate patients who may need hospitalization [44]. In our study, this value was found to be 8 points.

Corticosteroid use was associated with an increased risk of hospitalization [45,46]. These data could be interpreted in light of the fact that these drugs have been reserved, in our area, for patients who are on average more serious (hyperpyrexia resistant to NSAIDs or peripheral SpO2 <94% with the need for home oxygen therapy) [33]: this also highlights the inability of this therapeutic tool to recover most of the advanced clinical situations not yet in need for hospital treatment. A portion of the patients treated with corticosteroids may have started them in an untimely manner. As is known, these drugs promote viral replication; using them in the first week of the onset of the disease could fuel the replication of SARS-CoV-2, inducing greater lung damage in the second week of illness [31]. Our data show that the use of the steroid before the eighth day of illness is more associated with the need for hospitalization.

COVID-19 is a disease that is very variable in clinical aspects and courses; some symptom subsets can predict a less aggressive disease than others and provide useful information in the home management of patients with SARS-CoV-2. Likewise, LUS, MEWS, and the walking test can represent effective evaluation tools in identifying early clinical deterioration that requires hospital care, representing important weapons in the arsenal of the physician.

## 5. Conclusions

SARS-CoV-2 infection causes an aggressive, off-scale immune reaction that results in systemic and lung damage. The use of pidotimod proved able to prevent this phenomenon by directing the immune system toward a more appropriate and effective response, managing to prevent the evolution of the disease from mild-moderate to severe forms, which require hospitalization.

Furthermore, the use of pidotimod is able to speed up the negativization of the infection from SARS-CoV-2. Pidotimod is configured as an effective therapeutic option, inexpensive and usable in almost all categories of patients. Acting on the immune response may also represent a valid strategy in light of the emergence of new variants and resistance that can reduce the effectiveness of monoclonal antibodies and antivirals.

## Figures and Tables

**Figure 1 microorganisms-10-02131-f001:**
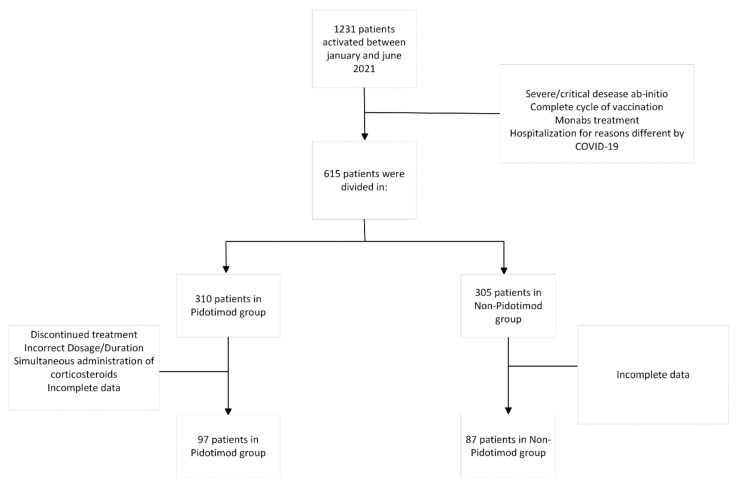
Study enrolment.

**Figure 2 microorganisms-10-02131-f002:**
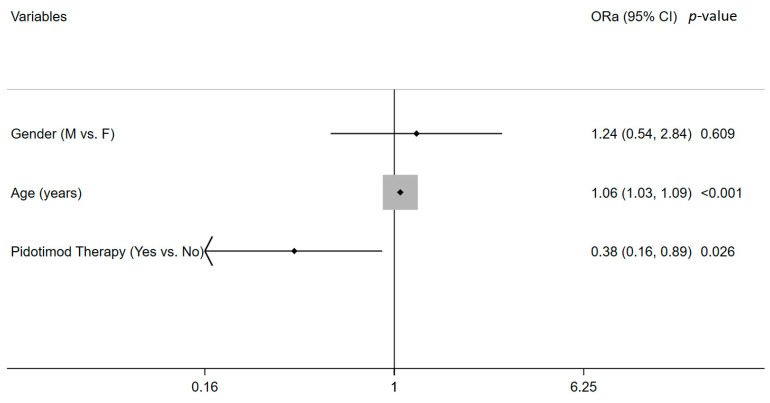
Logistic regression for ER access/hospitalization; Ora = adjusted odds ratio; 95% CI = 95% interval confidence.

**Table 1 microorganisms-10-02131-t001:** The patients’ socio-demographic and clinical characteristics (*n* = 184).

Gender	
F	100 (54.3%)
M	84 (45.7%)
Age (years)	53.0 (42.0–62.5)
Illness Duration (days)	22.0 (18.0–28.0)
Comorbidity	
No	59 (34.3%)
Yes	113 (65.7%)
BMI (kg/m^2^)	27.1 (23.8–29.4)
Fever	
No	51 (27.9%)
Yes	132 (72.1%)
Maximum Temperature	38.0 (37.6–38.5)
Days of Fever	3.0 (0.0–6.0)
Cough	
No	24 (13.0%)
Yes	160 (87.0%)
Dyspnea	
No	139 (75.5%)
Yes	45 (24.5%)
Ageusia or Anosmia	
No	88 (47.8%)
Yes	96 (52.2%)
Fatigue	
No	41 (22.3%)
Yes	143 (77.7%)
Myalgia	
No	64 (34.8%)
Yes	120 (65.2%)
Headache	
No	104 (56.5%)
*Yes*	80 (43.5%)
Others	
No	86 (47.3%)
Yes	96 (52.7%)
Number of Symptoms	4.0 (3.0–6.0)
SpO2 at Enrolment	98.0 (97.0–99.0)
Minimum SpO2	96.0 (94.0–97.0)
Mews	1.0 (0.0–3.0)
LUS	6.0 (2.0–10.0)
Walking test	97.0 (95.0–99.0)
Pidotimod Therapy	
No	87 (47.3%)
Yes	97 (52.7%)
Pidotimod Start Day	3.0 (2.0–6.0)
Pidotimod Duration	3.0 (0.0–10.0)
Corticosteroid Therapy	
No	119 (65.0%)
Yes	64 (35.0%)
Corticosteroid Start Day	8.0 (7.0–10.0)
ER Access/Hospitalization	
No	150 (81.5%)
Yes	34 (18.5%)

N (%) or median and IQR are shown when appropriate.

**Table 2 microorganisms-10-02131-t002:** The patients’ socio-demographic and clinical characteristics for pidotimod therapy (*n* = 184).

	Pidotimod Therapy		
	No (*n* = 87)	Yes (*n* = 97)	* *p*-Value
Gender			
F	47 (54.0%)	53 (54.6%)	0.526
M	40 (46.0%)	44 (45.4%)	
Age (years)	53.0 (44.0–64.0)	53.0 (40.0–61.0)	0.240
Illness Duration (days)	23.0 (20.0–31.0)	21.0 (17.0–27.0)	0.005
Comorbidity			
No	19 (24.4%)	40 (42.6%)	0.012
Yes	59 (75.6%)	54 (57.4%)	
BMI (kg/m^2^)	27.1 (26.0–29.4)	26.9 (23.5–29.4)	0.508
Fever during the course			
No	26 (29.9%)	25 (26.0%)	0.563
Yes	61 (70.1%)	71 (74.0%)	
Maximum Temperature	38.0 (37.5–38.5)	38.0 (37.6–38.5)	0.914
Days of Fever	3.0 (0.0–6.0)	3.0 (0.0–6.0)	0.438
Cough **			
No	14 (16.1%)	10 (10.3%)	0.245
Yes	73 (83.9%)	87 (89.7%)	
Dyspnea **			
No	64 (73.6%)	75 (77.3%)	0.753
At rest and in motion	11 (12.6%)	9 (9.3%)	
Only in motion	12 (13.8%)	13 (13.4%)	
Ageusia or Anosmia **			
No	46 (52.9%)	42 (43.3%)	0.194
Yes	41 (47.1%)	55 (56.7%)	
Fatigue **			
No	24 (27.6%)	17 (17.5%)	0.102
Yes	63 (72.4%)	80 (82.5%)	
Myalgia **			
No	37 (42.5%)	27 (27.8%)	0.037
Yes	50 (57.5%)	70 (72.2%)	
Headache **			
No	57 (65.5%)	47 (48.5%)	0.020
Yes	30 (34.5%)	50 (51.5%)	
Others **			
No	49 (56.3%)	37 (38.9%)	0.113
Yes	28 (43.7%)	57 (61.1%)	
Number of Symptoms **	4.0 (3.0–5.0)	5.0 (3.0–6.0)	<0.001
SpO2 at Enrolment	98.0 (96.0–99.0)	98.0 (97.0–99.0)	0.103
Minimum SpO2 during the course	95.0 (93.0–97.0)	96.0 (94.0–97.0)	0.046
Mews at Enrolment	1.0 (0.0–3.0)	1.0 (0.0–2.0)	0.062
Maximum Mews during the course	3.0 (0.0–5.0)	1.0 (0.0–3.0)	0.770
Maximum LUS during the course	7.0 (2.0–12.0)	4.5 (1.5–8.0)	0.128
Minimum SpO2 Walking Test	97.0 (93.0–98.0)	97.0 (96.0–99.0)	0.010
Corticosteroid Therapy			
No	34 (39.1%)	85 (88.5%)	<0.001
Yes	53 (60.9%)	11 (11.5%)	
Corticosteroid Start Day	8.0 (7.0–9.0)	8.0 (7.0–12.0)	0.282
Corticosteroid Duration	3.5 (0.0–11.0)	0.0 (0.0–0.0)	<0.001
ER Access/Hospitalization			
No	64 (73.6%)	86 (88.7%)	0.008
Yes	23 (26.4%)	11 (11.3%)	

N (%) or median (IQR) are shown when appropriate; * *p*-values are for the Wilcoxon rank-sum test or Pearson’s chi-square test; ** Symptoms present during the course of disease.

**Table 3 microorganisms-10-02131-t003:** The patients’ socio-demographic and clinical characteristics for ER access/hospitalization (*n* = 184).

	ER Access/Hospitalization		
	No (*n* = 150)	Yes (*n* = 34)	* *p*-Value
Gender			
F	84 (56.0%)	16 (47.1%)	0.225
M	66 (44.0%)	18 (52.9%)	
Age (years)	50.5 (39.0–60.0)	66.0 (54.0–79.0)	<0.001
Illness Duration (days)	22.0 (18.0–27.0)	28.0 (23.0–38.0)	0.002
Comorbidity			
No	55 (39.6%)	4 (12.1%)	0.003
Yes	84 (60.4%)	29 (87.9%)	
BMI (kg/m^2^)	26.2 (23.5–29.1)	33.3 (27.1–40.8)	0.010
Fever during the course			
No	46 (30.9%)	5 (14.7%)	0.058
Yes	103 (69.1%)	29 (85.3%)	
Maximum Temperature	38.0 (37.5–38.5)	38.2 (37.8–38.9)	0.183
Days of Fever	3.0 (0.0–5.0)	4.0 (2.0–7.0)	0.031
Cough **			
No	21 (14.0%)	3 (8.8%)	0.418
Yes	129 (86.0%)	31 (91.2%)	
Dyspnea **			
No	121 (80.7%)	18 (52.9%)	<0.001
At rest and in motion	9 (6.0%)	11 (32.4%)	
Only in motion	20 (13.3%)	5 (14.7%)	
Ageusia or Anosmia **			
No	65 (43.3%)	23 (67.6%)	0.010
Yes	85 (56.7%)	11 (32.4%)	
Fatigue **			
No	30 (20.0%)	11 (32.4%)	0.118
Yes	120 (80.0%)	23 (67.6%)	
Myalgia **			
No	49 (32.7%)	15 (44.1%)	0.206
Yes	101 (67.3%)	19 (55.9%)	
Headache **			
No	78 (52.0%)	26 (76.5%)	0.009
Yes	72 (48.0%)	8 (23.5%)	
Others **			
No	67 (45.3%)	19 (55.9%)	0.336
Yes	81 (54.7%)	21 (44.1%)	
Number of Symptoms **	4.0 (3.0–6.0)	4.0 (3.0–5.0)	0.127
SpO2 at Enrolment	98.0 (97.0–100.0)	96.0 (95.0–98.0)	<0.001
Minimum SpO2 during the course	96.0 (95.0–97.0)	93.0 (90.0–94.0)	<0.001
Maximum Mews during the course	1.0 (0.0–2.0)	5.0 (3.0–7.0)	0.001
Maximum LUS during the course	4.5 (1.5–6.5)	12.0 (8.0–14.0)	0.003
Minimum SpO2 at Walking Test	97.0 (95.0–99.0)	93.0 (90.5–97.0)	<0.001
Pidotimod Therapy			
No	64 (42.7%)	23 (67.6%)	0.008
Yes	86 (57.3%)	11 (32.4%)	
Pidotimod Start Day	3.0 (2.0–6.0)	4.0 (1.0–6.0)	0.792
Pidotimod Duration	7.0 (0.0–10.0)	0.0 (0.0–2.0)	<0.001
Corticosteroid Therapy			
No	102 (68.5%)	17 (50.0%)	0.042
Yes	47 (31.5%)	17 (50.0%)	
Corticosteroid Start Day	8.5 (8.0–11.0)	7.0 (4.5–8.0)	<0.001

N (%) or median (IQR) are shown when appropriate; * *p*-values are for the Wilcoxon rank-sum test or Pearson’s chi-square test; ** Symptoms present during the course of disease.

**Table 4 microorganisms-10-02131-t004:** Crude odds ratio and corresponding 95% CI for ER access/hospitalization.

	ORc	95%CI	*p*-Value
Gender (M vs. F)	1.431	[0.678, 3.020]	0.346
Age (years)	1.064	[1.036, 1.092]	<0.0001
BMI (kg/m^2^)	1.220	[1.059, 1.427]	0.007
Illness Duration (days)	1.073	[1.031, 1.116]	<0.0001
Comorbidity (Yes vs. No)	4.747	[1.581, 14.250]	0.005
Dyspnea (Yes vs. No)	1.648	[1.034, 2.624]	0.035
Ageusia or Anosmia (Yes vs. No)	0.365	[0.166, 0.804]	0.012
Headache (Yes vs. No)	0.333	[0.141, 0.783]	0.012
Pidotimod Therapy (Yes vs. No)	0.355	[0.161, 0.782]	0.010
Corticosteroid Therapy (Yes vs. No)	2.170	[1.019, 4.621]	0.045

ORc = crude odds ratio; 95% CI = 95% interval confidence.

**Table 5 microorganisms-10-02131-t005:** Multiple linear model for the illness duration.

	Coeff	95%CI	*p*-Value
Gender	−0.970	[−3.825, 1.879]	0.502
Age (years)	0.163	[0.078, 0.248]	<0.0001
Pidotimod Therapy (Yes)	−4.190	[−7.041, −1.340]	0.004

Coeff = coefficients; 95% CI = 95% interval confidence.

## Data Availability

Data available upon request.

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
