# Peer review of "New Therapeutic Options in Mild Moderate COVID-19 Outpatients"

_microorganisms, 2022, doi:10.3390/microorganisms10112131_

Round 1

Reviewer 1 Report

This  is an interesting well-written paper.

I think it could be accepted after minor revisions.

I think it could be helpful for the reader to report each IQR when you are describing your results.

Moreover, a complete revision of English should be performed.

Finally, I think in discussion Authors should better focus on the place in therapy of this drug (i.e. unvaccinated patients with no risk factors, young vaccinated patients) as at the moment antivirals demonstrated efficacy against all circulating VoCs.

See also:

- Real-world effectiveness of molnupiravir and nirmatrelvir/ritonavir among COVID-19 inpatients during Hong Kong’s Omicron BA.2 wave: an observational study  Nirmatrelvir/ritonavir and molnuipiravir in the treatment of mild/moderate COVID-19: results of a real-life study Early Clinical Experience with Molnupiravir for Mild to Moderate Breakthrough COVID-19 among Fully Vaccinated Patients at Risk for Disease Progression. Vaccines 202210, 1141. https://doi.org/10.3390/vaccines10071141   Oral Nirmatrelvir/Ritonavir Therapy for COVID-19: The Dawn in the Dark? Antibiotics 202211, 220. https://doi.org/10.3390/antibiotics11020220  

Author Response

  1. Reviewer

This is an interesting well-written paper.

I think it could be accepted after minor revisions.

we thank the reviewer for the comment

I think it could be helpful for the reader to report each IQR when you are describing your results.

we added IQR in the results as suggested

Moreover, a complete revision of English should be performed.

we have reviewed and corrected the text

Finally, I think in discussion Authors should better focus on the place in therapy of this drug (i.e. unvaccinated patients with no risk factors, young vaccinated patients) as at the moment antivirals demonstrated efficacy against all circulating VoCs.

See also:

- Real-world effectiveness of molnupiravir and nirmatrelvir/ritonavir among COVID-19 inpatients during Hong Kong’s Omicron BA.2 wave: an observational study  medRxiv 2022.05.19.22275291; doi:https://doi.org/10.1101/2022.05.19.22275291 -Nirmatrelvir/ritonavir and molnuipiravir in the treatment of mild/moderate COVID-19: results of a real-life study medRxiv 2022.08.23.22278585; doi:https://doi.org/10.1101/2022.08.23.22278585  - Early Clinical Experience with Molnupiravir for Mild to Moderate Breakthrough COVID-19 among Fully Vaccinated Patients at Risk for Disease Progression. Vaccines 202210, 1141. https://doi.org/10.3390/vaccines10071141 -   Oral Nirmatrelvir/Ritonavir Therapy for COVID-19: The Dawn in the Dark? Antibiotics 202211, 220. https://doi.org/10.3390/antibiotics11020220  

we revised the discussion, highlighting the role of the pidotimod better as suggested

Reviewer 2 Report

In this study, the authors analyzed the potential effect of Pidotimod use to prevent the disease progression in mildto moderate COVID-19 outpatients. As the Pidotimod's immunomodulatory activity and clinical effectiveness in respiratory tract infections, it is important to evaluate it's potential as a regular or standardized treatment for COVID19. There are several points needs to be addressed before publication:

How this study was designed? How did the authors determine the group of patient that received Pidotimod? 

Did they measure the SARS-CoV2 viral loads, what's the difference of VL between the two groups before Pidotimod use. Did the use of Pidotimod would reduce the VL?

How did the author explain several symptoms, such as Ageusia/Anosmia/headache, are associated with slow disease progerssion? Are these symptoms have also been observed in a large cohort or studies to be associted with slow progerssion or mild sysptoms? It would more convincing to evaluate the observations in another large cothort, which is also imporant to identify certain clinical scores/values (such as the use of SPO2 <94%, MEWS>2, LUS>8 in this study) that may be used as markers for physician decision.

Because Pidotimod has the immunomodulatory activity, are the immune cell counts different before and after the treatment?

Author Response

  1. Reviewer

In this study, the authors analysed the potential effect of Pidotimod use to prevent the disease progression in mild to moderate COVID-19 outpatients. As the Pidotimod's immunomodulatory activity and clinical effectiveness in respiratory tract infections, it is important to evaluate it's potential as a regular or standardized treatment for COVID19. There are several points needs to be addressed before publication:

How this study was designed? How did the authors determine the group of patient that received Pidotimod? 

We highlighted the study design in the materials and methods. Patients were retrospectively selected according to the inclusion criteria.  Pidotimod was administered by virtue of the presence of local guidelines that indicate pidotimod as therapeutic option in COVID patient’s

In our country, territorial management of patients affected by sars-cov-2 was done through special units (called u.s.c.a.) that visit and monitoring patients at home.

In October 2020 local health authority created territorial guidelines for COVID patients’ management outside the hospital (attached, in Italian language): this guideline contains different suggested therapies for treating territorial COVID patients, including Pidotimod.

The choice of COVID therapies was made by the USCA physicians based on the local guideline options.

Did they measure the SARS-CoV2 viral loads, what's the difference of VL between the two groups before Pidotimod use. Did the use of Pidotimod would reduce the VL?

VL were not measured. A PCR test was used to determine the presence of SARS COV2. We did not obtain data on the viraemia decay in the two groups, however the analysis shows a faster negativization time in patients of the pidotimod group. It is likely that the effect on the immune system and in particular on IFN gamma is responsible for these results ( “Pidotimod: the state of art,” Clin Mol Allergy, vol. 13, no. 1, May 2015)

How did the author explain several symptoms, such as Ageusia/Anosmia/headache, are associated with slow disease progerssion? Are these symptoms have also been observed in a large cohort or studies to be associted with slow progerssion or mild sysptoms? It would more convincing to evaluate the observations in another large cothort, which is also imporant to identify certain clinical scores/values (such as the use of SPO2 <94%, MEWS>2, LUS>8 in this study) that may be used as markers for physician decision.

During data analysis we observed that several symptoms are more prevalent in patients with lower disease progression. In literature is reported an association between this symptoms and between headache and a better outcome (“Factors associated with the presence of headache in hospitalized COVID-19 patients and impact on prognosis: a retrospective cohort study,” J Headache Pain, vol. 21, no. 1, Jul. 2020;  “The impact of headache disorders on COVID-19 survival: a world population-based analysis,” medRxiv, p. 2021.03.10.21253280, Mar. 2021;  “Analysis of clinical characteristics and outcomes in patients with COVID-19 based on a series of 1000 patients treated in Spanish emergency departments”). In the literature there are already evidences of the usefulness of MEWS  (Prognostic value of three rapid scoring scales and combined predictors for the assessment of patients with coronavirus disease 2019. Nursing Open (3), 1865–1872.; The utility of MEWS for predicting the mortality in the elderly adults with COVID-19: A retrospective cohort study with comparison to other predictive clinical scores), there is no agreement on the threshold to be used: our data shows that a value greater than 2 could be worth noting by the physician. A similar argument can be made with the LUS: in this case, other works suggest thresholds close to ours (“Role of Lung Ultrasound in the Management of Patients with Suspected SARS-CoV-2 Infection in the Emergency Department,” J Clin Med, vol. 11, no. 8, Apr. 2022; Ultrasonic Characteristics and Severity Assessment of Lung Ultrasound in COVID-19 Pneumonia in Wuhan, China: A Retrospective, Observational Study. Engineering (Beijing, China), 7(3), 367–375)

Because Pidotimod has the immunomodulatory activity, are the immune cell counts different before and after the treatment?

we did not count the immune cells before and after the treatment, because this was a study done with patients at home and there were no have a blood tests.

It is possible to believe that a rebalancing of the lymphocyte populations is obtained, as demonstrated in HIV subjects(Pidotimod and Immunological Activation in Individuals Infected with HIV. Curr HIV Res. 2021;19(3):260-268)

Round 2

Reviewer 2 Report

I have no further questions.